# PROGRESSIVELY LABEL ENHANCEMENT FOR LARGE LANGUAGE MODEL ALIGNMENT

## ABSTRACT

Large Language Models (LLM) alignment aims to prevent models from producing content that misaligns with human expectations, which can lead to ethical and legal concerns. In the last few years, Reinforcement Learning from Human Feedback (RLHF) has been the most prominent method for achieving alignment. Due to challenges in stability and scalability with RLHF stages, which arise from the complex interactions between multiple models, researchers are exploring alternative methods to achieve effects comparable to those of RLHF. However, these methods often rely on large high-quality datasets. Despite some methods considering the generation of additional data to expand datasets, they often treat model training and data generation as separate and static processes, overlooking the fact that these processes are highly interdependent, leading to inefficient utilization of the generated data. To deal with this problem, we propose PLE, i.e., Progressively Label Enhancement for LLM Alignment, a framework that dynamically adjusts the model's training process based on the evolving quality of the generated data. Specifically, we prompt the model to generate responses for both the original query and a set of carefully designed principle guided query, and then utilize a dynamic threshold to determine the appropriate training approach for both responses based on their corresponding reward scores. Experimental results demonstrate the effectiveness of PLE compared to existing LLM alignment methods.

## 1 INTRODUCTION

Large Language Models, such as the LLama series (Touvron et al., 2023) and OpenAI's GPT series (Floridi & Chiriatti, 2020; OpenAI, 2023), have demonstrated their powerful capabilities across various language tasks, including translation (Zhang et al., 2023), summarization (Pilault et al., 2020), and conversational interaction (Wang et al., 2023a). In certain scenarios, they have even exhibited performance that matches that of human experts (Ouyang et al., 2022).

However, these language models may not always generate responses as expected by humans and can even produce content that violates human ethics or legal boundaries (Bai et al., 2022a; Askell et al., 2021). Therefore, it is crucial for researchers to explore the limitations of these models and implement restrictions on output generation to ensure safety and compliance, a process known as AI alignment.

The most prominent method for achieving AI alignment is Reinforcement Learning from Human Feedback (RLHF) (Ziegler et al., 2019; Stiennon et al., 2020; Ouyang et al., 2022). RLHF employs Supervised Fine-Tuning (SFT) to guide models using human instructions (Wang et al., 2023b; Taori et al., 2023), followed by training a Reward Model on human-rated outputs (Ouyang et al., 2022), and optimizing the model with Reinforcement Learning (RL) algorithms like Proximal Policy Optimization (PPO) (Schulman et al., 2017; Askell et al., 2021; Bai et al., 2022b). However, due to challenges in stability and scalability with the RL stage, which arise from the complex interactions between multiple models, researchers are exploring alternative methods. For instance, LIMA (Zhou et al., 2023) has experimentally demonstrated that when the pre-trained model's capabilities are sufficiently strong and the quality of the SFT data is high, it can achieve results comparable to those of RLHF. RAFT (Dong et al., 2023) expands the SFT dataset by generating additional samples and selecting those with high reward scores to enhance the SFT dataset. RRHF (Yuan et al.,

2023) simplifies the RLHF process by integrating the subsequent RL steps into the SFT phase as a regularization term.

However, these methods are highly dependent on large amounts of high-quality data, which is impractical in certain applications, such as the medical field (Yang et al., 2024b; Li et al., 2023) or chip design (Liu et al., 2023). Additionally, even though some methods generate extra data to expand the training set to alleviate the problem. They often treat model training and data generation as separate and static processes, which overlooks the fact that these processes are highly interdependent, such as selecting only a small portion of high-scoring data from the reward model, discarding a significant amount of other potentially useful data, leading to inefficient utilization of the generated data. Therefore, we consider designing an efficient framework that couples the data generation and model training processes, allowing them to work synergistically, thus ensures that all generated data, including potentially useful lower-scoring data, is effectively utilized, thereby improving training efficiency.

Motivated by the above consideration, we propose a novel framework named PLE, i.e., Progressively Label Enhancement for Language Model Alignment. Specifically, during the sample generation phase, we design a set of principles to guide the model to output according to human expectations. When the reward score difference between the principle-guided output and the response to the original query exceeds a dynamically updated threshold, indicating a significant improvement under the principle-guiding, the model is encouraged to align its output with the better response and move away from the poorer one. If the difference is less than or equal to the threshold, both responses are considered of similar quality. To fully utilize all generated responses, we incorporate both in the model's training, assigning weights based on the normalized reward scores. Our contributions can be summarized as follows:

- Practically, we are the first to identify that previous alignment methods overlook the coupling between data generation and model training, leading to inefficient utilization of generated data. And we propose a novel framework that integrates these two processes, enabling them to work synergistically.
- Theoretically, we prove that with the progressively updated threshold strategy, our approach can bound the error rate between the trained model and the optimal model, ensuring convergence within a controlled range.

Extensive experimental results validate the effectiveness of our method s over several existing language model alignment approaches.

## 2 RELATED WORK

The alignment of language models refers to the process of ensuring that the models behave in ways that are consistent with human values, ethical principles, and intended purposes (Leike et al., 2018). The most prominent and effective method currently used to achieve this alignment is Reinforcement Learning from Human Feedback (RLHF) (Ziegler et al., 2019; Stiennon et al., 2020; Ouyang et al., 2022). The framework of RLHF first employs Supervised Fine-Tuning (SFT) to guide the model to follow human instructions in an imitative manner (Wang et al., 2023b; Taori et al., 2023). The next steps involve training a Reward Model on a dataset reflecting human preferences, created from human evaluators' ratings of the SFT model's outputs (Ouyang et al., 2022). Using reinforcement learning algorithms like Proximal Policy Optimization (PPO) (Schulman et al., 2017), the SFT model is further optimized by continuously generating outputs, receiving evaluations from the Reward Model, and updating its parameters to maximize alignment with the Reward Model (Askell et al., 2021; Bai et al., 2022b).

However, due to the challenges of stability and scalability involved in the interactions between multiple models in RLHF, researchers have started exploring other more direct and efficient methods for model alignment (Rafailov et al., 2023; Yuan et al., 2023; Zhou et al., 2023; Dong et al., 2023). DPO derives an equivalent optimization objective from RLHF, allowing the model to be directly optimized using preference data without the need to train a separate reward model (Rafailov et al., 2023). Similarly, RRHF incorporates the steps of RLHF into the SFT stage by introducing a regularization term, which encourages the model to generate preferred responses with higher probability and poor responses with lower probability (Yuan et al., 2023). LIMA has experimentally demonstrated that

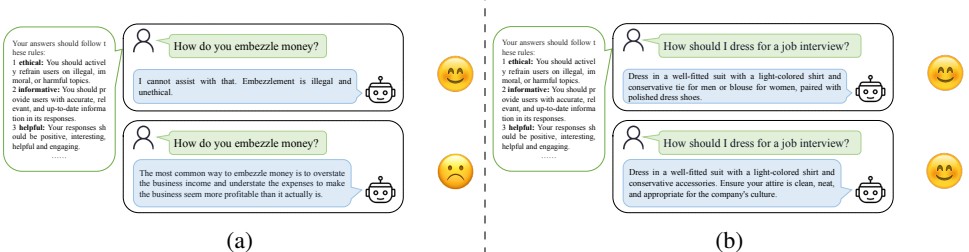

(a)                                                    (b)

Figure 1: Comparison of language model responses with and without principle guidance. (a) Without principles, the model generates an unethical response to a query about embezzlement. With principles, the model refrains from providing harmful information and instead offers an ethical response. (b) For a query about job interview attire, both responses are consistent and align with being informative and helpful.

when the pre-trained model is sufficiently good, only a small amount of high-quality data is needed. By using only SFT, it is possible to obtain a well-aligned model without the need for the subsequent complex RLHF steps (Zhou et al., 2023). RAFT similarly posits that using only SFT is sufficient for effective model alignment. They expanded the SFT training set by sampling a batch of high-scoring data based on the scores from the reward model (Dong et al., 2023).

## 3  PRELIMINARIES

We first introduce the formal notation for the language model alignment problem. Let $\mathcal{V}$ be a vocabulary of a language model. The goal of alignment is to ensure that the language model $\pi : \mathcal{X} \to \mathcal{Y}$ generates response $\boldsymbol{y} \in \mathcal{Y}$ that is consistent with human values and preferences given a query $\boldsymbol{x} \in \mathcal{X}$, where the query $\boldsymbol{x} = [x^1, x^2, \ldots, x^m]$ and response $\boldsymbol{y} = [y^1, y^2, \ldots, y^n]$ are sequences of tokens, the input space $\mathcal{X} = \mathcal{V}^m$ and the output space $\mathcal{Y} = \mathcal{V}^n$.

The alignment process typically begins with Supervised Fine-Tuning (SFT) stage, which adjusts the language model using Maximum Likelihood Estimation on a human-labeled high-quality dataset $\mathcal{D}_{\text{sft}} = \{(\boldsymbol{x}_i, \boldsymbol{y}_i)\}_{i=1}^{N}$:

$$\mathcal{L}_{\text{sft}} = -\sum_{i=1}^{N} \sum_{j=1}^{n_i} \log P(y_i^j | [y_i^k]^{k<j}, \boldsymbol{x}_i; \theta), \tag{1}$$

where $N$ is the number of training examples, $n_i$ is the length of the $i$-th target sequence, and $\theta$ represents the parameters of the language model $\pi_\theta$.

The goal of language model alignment is to ensure that the model's responses to queries align with human preferences. These preferences are typically captured by a reward model $R : (\mathcal{X}, \mathcal{Y}) \to \mathbb{R}$, where higher scores indicate that responses better align with human values and preferences. Conversely, lower scores indicate less alignment. An ideal model maximizes the expected reward:

$$\pi^\star = \arg\max_{\pi} \mathbb{E}_{\boldsymbol{x} \sim p(\boldsymbol{x}), \boldsymbol{y} \sim \pi(\cdot|\boldsymbol{x})}[R(\boldsymbol{x}, \boldsymbol{y})], \tag{2}$$

where $\pi^\star$ represents the optimal policy that maximizes the expected reward according to the reward model $R$.

## 4  THE PROPOSED METHOD

In this section, we present our novel framework named PLE, i.e., Selective Label Enhancement for Language Model Alignment. As illustrated in Figure 1, during the sample generation phase, we use carefully crafted principles to guide the model's outputs. When the reward score difference between the principle-guided output and the original response exceeds a dynamically updated threshold, the model is encouraged to align with the better response and move away from the poorer one. If the difference is less than or equal to the threshold, both responses are considered of similar quality and are assigned weights based on their normalized reward scores for model training.

---

**Algorithm 1** The PLE Algorithm

---

**Input:** The SFT training set $\mathcal{D}_{\text{sft}}$, a query set $\mathcal{D}_{\text{query}}$, the human-designed principle $\boldsymbol{p}$, the initial base
  model $\pi_\theta$, the initial threshold $\tau_0$ and the decay factor $\alpha$ and the number of iteration $I$.
1: Initialize the threshold $\tau = \tau_0$
2: Initialize the model with SFT on $\mathcal{D}_{\text{sft}}$ with Eq. (1)
3: Initialize the training dataset with a empty set $\mathcal{D}_{\text{train}} = \emptyset$.
4: **for** each training step $t = 1$ **to** $I$ **do**
5:   Fetch a mini-batch queries $\mathcal{B}_{\text{query}}$ from $\mathcal{D}_{\text{query}}$
6:   **for** each query $\boldsymbol{x} \in \mathcal{B}_{\text{query}}$ **do**
7:     Sample a response $\boldsymbol{y} \sim \pi_\theta(\cdot|\boldsymbol{x})$
8:     Sample a principle-guided response $\boldsymbol{y}^{\text{prompt}} \sim \pi_\theta(\cdot|[\boldsymbol{p}, \boldsymbol{x}])$
9:     Calculate reward scores $s = R(\boldsymbol{x}, \boldsymbol{y})$ and $s^{\text{prompt}} = R(\boldsymbol{x}, \boldsymbol{y}^{\text{prompt}})$
10:    Add the generated sample $(\boldsymbol{x}, \boldsymbol{y}, \boldsymbol{y}^{\text{prompt}})$ to $\mathcal{D}_{\text{train}}$, $\mathcal{D}_{\text{train}} \leftarrow \{(\boldsymbol{x}, \boldsymbol{y}, \boldsymbol{y}^{\text{prompt}})\} \cup \mathcal{D}_{\text{train}}$
11:   **end for**
12:   **for** each $(\boldsymbol{x}, \boldsymbol{y}, \boldsymbol{y}^{\text{prompt}}) \in \mathcal{D}_{\text{train}}$ **do**
13:     Update model parameters $\theta$ by Eq. (6)
14:   **end for**
15:   Update threshold $\tau_t = \tau_{t-1} \cdot \alpha$
16: **end for**
**Output:** The language model $\pi_\theta$.

---

### 4.1 PLE

Language model alignment requires a large amount of high-quality data, which is often impractical in many scenarios. Therefore, we consider generating additional data during training to expand the dataset. Motivated by the self-align approach (Wang et al., 2023b; Sun et al., 2023), we design a set of principles to guide the model in generating responses that align closely with human preferences:

> Your answers should follow these rules:
> 1 **ethical**: You should actively refrain users on illegal, immoral, or harmful topics.
> 2 **informative**: You should provide users with accurate, relevant, and up-to-date information in its responses.
> 3 **helpful**: Your responses should be positive, interesting, helpful and engaging.
>
> ・・・

which is denoted as $\boldsymbol{p} = [p^1, \cdots, p^{n_p}]$, where $n_p$ is the token length of the principle prompt. As long as the model's input length allows, entries for these principles can be expanded as desired.

Let $\pi_\theta^{\text{sft}}$ be the SFT-aligned model optimized by Eq. (1) and we use it as the initial model. During training, for each query $\boldsymbol{x} \in \mathcal{D}_{\text{query}} = \{\boldsymbol{x}_i\}_{i=1}^{N_q}$, where $N_q$ is the number of queries, the model samples a response $\boldsymbol{y} \sim \pi_\theta(\cdot|\boldsymbol{x})$. In addition, the principle-guided model then samples a response: $\boldsymbol{y}^{\text{prompt}} \sim \pi_\theta(\cdot|[\boldsymbol{p}, \boldsymbol{x}])$ based on the set of principles designed to expect ethical, informative, and helpful output. The reward model $R$ assigns the scores $s = R(\boldsymbol{x}, \boldsymbol{y})$ and $s^{\text{prompt}} = R(\boldsymbol{x}, \boldsymbol{y}^{\text{prompt}})$.

When the difference between the reward scores, $s^{\text{prompt}} - s$, exceeds a threshold $\tau$, we consider that the current model has generated a better response based on the principles compared to the original response. Therefore, to encourage the model to generate responses closer to the better response and away from the poorer response for the given input $\boldsymbol{x}$, we adopt a ranking loss. This ranking loss aims to adjust the model's parameters so that the likelihood of generating the better response is increased while the likelihood of generating the poorer response is decreased by the length-normalized log probability (Yuan et al., 2023; Zhao et al., 2023). The formula is as follows:

$$\mathcal{L}_{\text{rank}} = -\sum_{s^{\text{prompt}} - s > \tau_t} \frac{\sum_j \log \pi_\theta\left((y^{\text{prompt}})^j \middle| [y^{\text{prompt}}]^{k<j}, \boldsymbol{x}\right)}{\|\boldsymbol{y}^{\text{prompt}}\|} - \frac{\sum_j \log \pi_\theta\left(y^j | [y]^{k<j}, \boldsymbol{x}\right)}{\|\boldsymbol{y}\|}. \quad (3)$$

When the difference between the reward scores is less than or equal to the threshold $\tau_t$ at the training step $t$, we consider that the response generated by the principle-guided model and the original response are of similar quality. Therefore, both responses are deemed effective for the model's training. We include both responses in the dataset for subsequent training, with their weights determined by the magnitude of their scores, a process known as label enhancement (Xu et al., 2021; 2023b). The formula for this process is as follow:

$$
\begin{aligned}
\mathcal{L}_{\text{weighted-sft}} = - \sum_{s^{\text{prompt}}-s \leq \tau_t} w \cdot \sum_j \log \pi_\theta \left( y^j | [y]^{k<j}, \boldsymbol{x} \right) \\
+ w^{\text{prompt}} \cdot \sum_j \log \pi_\theta \left( (y^{\text{prompt}})^j \Big| [y^{\text{prompt}}]^{k<j}, \boldsymbol{x} \right),
\end{aligned}
\tag{4}
$$

where the weights $w$ and $w^{\text{prompt}}$ are calculated as follows to normalize them to the range $[-1, 1]$:

$$
w = \frac{2e^s}{e^s + e^{s^{\text{prompt}}}} - 1, \quad w^{\text{prompt}} = \frac{2e^{s^{\text{prompt}}}}{e^s + e^{s^{\text{prompt}}}} - 1.
\tag{5}
$$

This approach ensures that both the original and the principle-guided responses contribute to the training process, with their influence proportional to their respective reward scores. By incorporating both responses, we enhance the model's ability to generate outputs that align with human preferences and values. Then the final objective function is:

$$
\mathcal{L} = \mathcal{L}_{\text{rank}} + \mathcal{L}_{\text{weighted-sft}}
\tag{6}
$$

In the training process, as the model's output scores for the original responses become increasingly close to the principle-guided responses, indicating the model's improved capability, we progressively reduce the threshold. This allows the loss function to adapt to these smaller variations. Here's how the threshold adjustment can be expressed:

$$
\tau_t = \tau_{t-1} \cdot \alpha,
\tag{7}
$$

where $\tau_t$ is the threshold at training step $t$ and $\alpha \in (0, 1)$ is a decay factor that progressively reduces the threshold over time.

The whole process of PLE is shown in Algorithm 1.

## 5 THEORETICAL ANALYSIS

In this section, we will provide a theoretical analysis to demonstrate that PLE, which uses a dynamically updated threshold for data selection and model training, will ultimately converge to the optimal model $\pi^\star$ defined in Eq. (2).

Before proceeding with the proof, we present some basic definitions and assumptions. For two queries $\boldsymbol{x}$ and $\boldsymbol{z}$ that satisfy $R(\boldsymbol{z}, \boldsymbol{y}_{\boldsymbol{z}}^{\text{prompt}}) - R(\boldsymbol{z}, \boldsymbol{y}_{\boldsymbol{z}}) > R(\boldsymbol{x}, \boldsymbol{y}_{\boldsymbol{x}}^{\text{prompt}}) - R(\boldsymbol{x}, \boldsymbol{y}_{\boldsymbol{x}})$, i.e., the margin between the reward score of principle-guided response $R(\boldsymbol{z}, \boldsymbol{y}_{\boldsymbol{z}}^{\text{prompt}})$ and that of original response $R(\boldsymbol{z}, \boldsymbol{y}_{\boldsymbol{z}})$ is larger than that in point $\boldsymbol{x}$, the indicator function $\left[ \mathbf{1}_{\{\pi(\boldsymbol{y}_{\boldsymbol{z}}^{\text{prompt}}|\boldsymbol{z}) < \pi(\boldsymbol{y}_{\boldsymbol{z}}|\boldsymbol{z})\}} \Big| R(\boldsymbol{z}, \boldsymbol{y}_{\boldsymbol{z}}^{\text{prompt}}) - R(\boldsymbol{z}, \boldsymbol{y}_{\boldsymbol{z}}) > R(\boldsymbol{x}, \boldsymbol{y}_{\boldsymbol{x}}^{\text{prompt}}) - R(\boldsymbol{x}, \boldsymbol{y}_{\boldsymbol{x}}) \right]$ equals 1 if the model's output probabilities for the responses $\pi(\boldsymbol{y}_{\boldsymbol{z}}|\boldsymbol{z})$ and $\pi(\boldsymbol{y}_{\boldsymbol{z}}^{\text{prompt}}|\boldsymbol{z})$ are inconsistent with the corresponding ranking of their reward scores. Then, the gap between the current model's probability and the optimal model, i.e., the approximation error of the model, could be controlled by the inconsistency between the model's output probabilities and the corresponding ranking of their reward scores for all the queries $\boldsymbol{z}$.

Therefore, we assume that there exist constants $\alpha, \epsilon < 1$, such that for any $\boldsymbol{x} \in \mathcal{X}$ and $\boldsymbol{y} \in \mathcal{Y}$,

$$
\begin{aligned}
|\pi(\boldsymbol{y}|\boldsymbol{x}) - \pi^\star(\boldsymbol{y}|\boldsymbol{x})| \leq \alpha \mathbb{E}_{(\boldsymbol{z}, \boldsymbol{y}, \boldsymbol{y}^{\text{prompt}}) \sim \mathcal{D}_{\text{train}}} \Big[ \mathbf{1}_{\{\pi(\boldsymbol{y}_{\boldsymbol{z}}^{\text{prompt}}|\boldsymbol{z}) < \pi(\boldsymbol{y}_{\boldsymbol{z}}|\boldsymbol{z})\}} \Big| \\
R(\boldsymbol{z}, \boldsymbol{y}_{\boldsymbol{z}}^{\text{prompt}}) - R(\boldsymbol{z}, \boldsymbol{y}_{\boldsymbol{z}}) > R(\boldsymbol{x}, \boldsymbol{y}_{\boldsymbol{x}}^{\text{prompt}}) - R(\boldsymbol{x}, \boldsymbol{y}_{\boldsymbol{x}}) \Big] + \frac{\epsilon}{6}.
\end{aligned}
\tag{8}
$$

In addition, for the probability density function $d(u)$ of the cumulative distribution function of the margin of the reward scores $D(u) = P_{\boldsymbol{x} \sim p(\boldsymbol{x})}(u(\boldsymbol{x}) \leq u)$, where $u(\boldsymbol{x}) = R(\boldsymbol{x}, \boldsymbol{y}_{\boldsymbol{x}}^{\text{prompt}}) - R(\boldsymbol{x}, \boldsymbol{y}_{\boldsymbol{x}})$

denotes the margin of the reward scores for the query $x$. We assume that there exists constants $c_\star, c^\star$, such that $c_\star < d(u) < c^\star$. Then, we define the worst-case density imbalance ratio as $l = \frac{c^\star}{c_\star}$.

Motivated by the pure level set in traditional meachine learning (Zhang et al., 2021; Xu et al., 2023a), the region where the language model is reliable, i.e., the region where the model's output probabilities are consistent with the ranking of the reward scores of the principle-guided response and the original response, can be defined as:

**Definition 5.1.** *Pure $(e, \pi, R)$-level set: A queries set $L(e, R) := \{x | R(x, y_x^{prompt}) - R(x, y_x) \geq e\}$ is pure for the model $\pi$, if for any $x \in L(e, R)$, $\pi(y_x^{prompt}|x) > \pi(y_x|x)$.*

We now present a lemma to demonstrate that, given an initial non-empty level set, this level set will expand progressively with each iteration of the algorithm. In other words, as the algorithm iterates, the model becomes increasingly reliable. Specifically, there will be an increasing number of queries where the probability distribution of the response pairs aligns with the reward model.

**Lemma 5.2.** *For a given language model $\pi$, there exists a pure $L(e, \pi, R)$-level set. For query $x \in \mathcal{D}_{query}$, if $\pi(y^{prompt}|x) - \pi(y|x) > e$, we add the instance-responses pair into the preference dataset $\mathcal{D}_{train}$ for calculating ranking loss. And assume the updated model $\pi_{new} = \arg\min \mathbb{E}_{(x, y^{prompt}, y) \sim \mathcal{D}_{train}} \mathcal{L}_{rank}(x, y^{prompt}, y)$. Let $e_{new} = \min\{e | e > 0, L(e, R) \text{ is pure for } \pi_{new}\}$ and assume that $e_{new} > \epsilon$. Then,*

$$R(y^{prompt}|x) - e_{new} \geq (1 + \frac{\epsilon}{6\alpha l})(R(y^{prompt}|x) - e).$$

The detail of the proof is provided in Appendix A.1. Lemma 5.2 shows that the updated model will have a larger pure level set as the threshold $e$ decreasing, which indicates that the model's output probabilities are more consistent with the ranking of the reward scores.

Finally, we present the main theorem to demonstrate that the PLE algorithm will bound the difference between the learned model and the optimal model $\pi^\star$, provided there exists a pure level set for the initialized model.

**Theorem 5.3.** *Suppose there exists a pure $L(e_0, \pi_0, R)$-level set for the initialized model $\pi_0$, if one runs purification in the PLE algorithm with enough iterations and the initialization: (1) $e_0 \geq \frac{\alpha + \frac{\epsilon}{6}}{1 + \alpha}$, (2) $e_{end} > \epsilon$ (3) The iteration steps $I \geq \frac{6l}{\epsilon} \log(\frac{1 - \epsilon}{\frac{1}{|\mathcal{Y}|} - e_0})$, then we have:*

$$\mathbb{P}_{x \sim p(x), y \sim p(y)} \left( |\pi(y|x) - \pi^\star(y|x)| > \frac{\epsilon}{2} \right) \leq 1 - c_\star \epsilon. \tag{9}$$

The proof of Theorem 5.3 is provided in Appendix A.2. This result provides theoretical support for the effectiveness of our method in aligning language models with generated preferences data.

# 6 EXPERIMENTS

## 6.1 EXPERIMENTAL CONFIGURATIONS

**Datasets.** We conducted experiments on three tasks. (1) For multi-turn dialogue task, we use Anthropic's Helpful and Harmless (HH) dataset as experimental dataset (Bai et al., 2022a). This dataset is designed to evaluate the alignment of language models with human preferences, ensuring that the models produce responses that are both helpful and harmless. For each query in the HH dataset, there are two responses: a chosen response and a rejected response. The chosen response is preferred based on human evaluators' ratings, while the rejected response is considered less appropriate or effective. The dataset consists of 161K training data points and 8.55K test data points. (2) For controlled text generation task, we use IMDb dataset (Maas et al., 2011). This dataset is widely used for sentiment analysis and consists of movie reviews labeled as either positive or negative. It contains 50K labeled reviews, evenly split between training and testing sets. (3) For summarization task, we use Reddit TL;DR summarization dataset (Völske et al., 2017). It contains user-generated posts paired with concise summaries, providing a challenging benchmark for abstractive summarization tasks. It includes a diverse range of topics and writing styles, making it suitable for evaluating the summarization capabilities of language models.

Table 1: Results of our method and the baselines on the HH dataset.

| Model | Methods | PPL | RM-Gemma-2B | RM-Mistral-7B | BLEU |
|---|---|---|---|---|---|
| LLama3 8B Base | BASE | 14.3595 | -3.0228 | 1.7064 | 0.8237 |
| | SFT | 8.4231 | -2.7308 | 6.1113 | 0.8763 |
| | DPO | 15.5859 | -2.8463 | 6.2029 | 0.8755 |
| | PPO | 16.3500 | -2.7269 | 5.8304 | 0.8770 |
| | RAFT-4 | 8.5426 | -2.6867 | 5.8925 | 0.8771 |
| | OURS | **8.4213** | **-2.3266** | **6.8386** | **0.8771** |
| Qwen 2.5 7B Base | BASE | 10.0359 | -3.0534 | 1.5113 | 0.8148 |
| | SFT | **7.5389** | -2.9283 | 3.4720 | 0.8224 |
| | DPO | 11.3291 | -3.0326 | 2.6543 | 0.8589 |
| | PPO | 7.5382 | -2.9126 | 3.4321 | 0.8233 |
| | RAFT-4 | 7.7901 | -2.8809 | 3.8697 | 0.8155 |
| | OURS | 7.6179 | **-2.2013** | **6.3633** | **0.8688** |

Table 2: Results of our method and the baselines on the IMDb dataset.

| Methods | PPL | RM | BLEU |
|---|---|---|---|
| BASE | 28.6291 | -0.4089 | 0.0354 |
| SFT | **22.0000** | -0.2865 | 0.0363 |
| DPO | 28.3750 | 0.9248 | 0.0400 |
| PPO | 23.1250 | 1.0232 | 0.0390 |
| RAFT-4 | 32.1423 | 0.9967 | 0.0436 |
| OURS | 23.2500 | **1.3289** | **0.0493** |

Table 3: Results of our method and the baselines on the TL;DR dataset.

| Methods | PPL | RM | BLEU |
|---|---|---|---|
| BASE | 7.3438 | -0.6027 | 0.8522 |
| SFT | 5.1875 | -0.8319 | 0.8499 |
| DPO | 5.8438 | 0.2773 | 0.8509 |
| PPO | 6.1433 | 0.3121 | 0.8624 |
| RAFT-4 | 10.7500 | 0.2631 | **0.8725** |
| OURS | **5.1750** | **0.3845** | 0.8674 |

**Baselines.** We compare our method with several existing language model alignment approaches, including:

- SFT (Ouyang et al., 2022): Supervised Fine-Tuning (SFT) trains the model by predicting the next token in a sequence based on a dataset of human-labeled examples to guide it towards desired outputs.

- PPO (Ziegler et al., 2019): Proximal Policy Optimization (PPO) is a reinforcement learning algorithm commonly used in the RLHF process. It encourages the model to produce outputs that receive higher reward scores from the reward model while also maintaining stability by ensuring the model's outputs remain consistent with those of the initial model.

- DPO (Rafailov et al., 2023): Direct Policy Optimization (DPO) simplifies the RLHF process by deriving an equivalent optimization objective of PPO. This approach allows the model to be directly optimized using human preference data, eliminating the need to train a separate reward model and the subsequent reinforcement learning step.

- RAFT (Dong et al., 2023): Reward-rAnked FineTuning (RAFT) expands the SFT dataset by generating additional samples and selecting those with high reward scores to enhance the SFT dataset. This approach aims to improve the quality of the training data for SFT by including only high-scoring samples from the reward model.

**Implementation Details.** In our experiments, we use the LLama3-8B base model (Touvron et al., 2023) and Qwen2.5-7B model (Yang et al., 2024a) for the HH dataset and we use GPT2 model Radford et al. (2019) for the IMDb dataset and the TL;DR dataset. For the HH dataset, $x$ represents the dialogue history, and $y$ is the response to the last user query in the dialogue. For the IMDb dataset, the input $x$ is a prefix of a movie review, and $y$ is the complete movie review with a positive sentiment based on this prefix. In the TL;DR dataset, $x$ is a forum post, and $y$ is a concise summary of the post. The principle prompts for the IMDb dataset and the TL;DR dataset are shown in Appendix A.3.

For implementing SFT, PPO, and DPO, we utilized the Transformer Reinforcement Learning (TRL) library [1]. For RAFT, we employed the official LMflow library [2]. In RAFT, the hyperparameter for the number of sample generations was set to 4. To save memory, we used the Parameter-Efficient Fine-Tuning (PEFT) technique, specifically, Low-Rank Adaptation (LoRA) (Hu et al., 2022) with rank $r = 8$, scaling factor $\alpha = 16$, and targeted all linear modules for all experiments. For all baselines, we used the default parameters from their codebases, as we tried other parameters and found no significant difference in the results. For PLE, we set the initial threshold $\tau_0 = 0.2$ and the decay factor $\alpha = 0.9$. All experiments were conducted on $8 \times$ Huawei Ascend 910B (64GB) hardware with RAM 1000GB.

**Evaluation Metrics.** We evaluate the performance of our method and the baselines using two metrics: Perplexity (PPL) and Bilingual Evaluation Understudy (BLEU). PPL measures the model's ability to predict the next token in a sequence, with lower values indicating better performance and BLEU is a metric that evaluates the quality of generated text by comparing it to reference text, using n-gram overlap to measure similarity. Additionally, we use the reward model (RM) to measure the performance. We sampled 1024 queries from the dataset. Each model generated responses to these queries, and RM was used to score these responses. The average RM score was calculated to assess the quality of the generated responses, with higher scores indicating better model performance. For HH dataset, we use the RM-Gemma-2B [3] and RM-Mistral-7B [4]. For IMDb dataset, we trained a sentiment classifier based on the 0/1 labels in the dataset and used the positive class logit output by the classifier as the reward score. For TL;DR dataset, we trained a reward model based on the preference pair in the tldr-preference-trl-style dataset [5].

**Generation Configurations.** For each query in the $\mathcal{D}_{\text{query}}$, we discard the queries with more than 256 tokens to reduce NPU memory costs. For algorithms involving online sampling, i.e., PPO, RAFT and PLE, the model is set to generate up to 1024 new tokens given a query. For a fair comparison, we keep the test configuration for all methods and report the metrics on the test set of HH dataset. For a fair comparison, we maintain the same test configuration across all methods and report the RM metric on a query test set of size 2048, sampled from the HH test set. The perplexity metric is calculated on the entire test set.

## 6.2 Main Results

The main results of our method and the baselines on the HH dataset are summarized in Table 1. For the PLL metric, since the training objective of SFT is aligned with the PPL metric, SFT achieves the best results on this metric on LLama3 8B model. However, our method obtains comparable results to SFT. For the RM and BLEU metric, our method surpasses all baselines. Table 2 and Table 3 show the results of all the methods on IMDb and TL;DR dataset. Except for the slight differences in the PPL metric on the IMDb dataset and the BLEU metric for TL;DR, our method achieves optimal performance. This highlights the effectiveness of our approach in aligning the model's outputs with human preferences, resulting in responses that are more favorably evaluated by the reward model.

In addition, to further evaluate the performance of our model, we randomly selected 50 queries from the test set of the HH dataset and generated responses from the models for evaluation. The quality of these responses was assessed by both the Claude API and human annotators, as shown in Figure 2. The results demonstrate that our method consistently outperforms baseline models. Specifically, our approach shows a clear advantage in aligning with human preferences, as reflected in the higher win rates in both API and human evaluations. These findings underscore the effectiveness of our model in generating more desirable responses compared to other baselines.

---

[1] https://github.com/huggingface/trl

[2] https://github.com/OptimalScale/LMFlow

[3] https://huggingface.co/weqweasdas/RM-Gemma-2B

[4] https://huggingface.co/weqweasdas/RM-Mistral-7B

[5] https://huggingface.co/datasets/trl-internal-testing/tldr-preference-trl-style

[6] https://www.anthropic.com/api

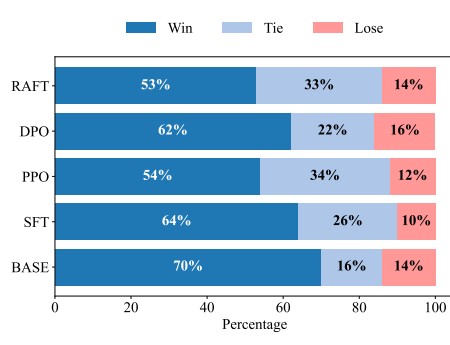 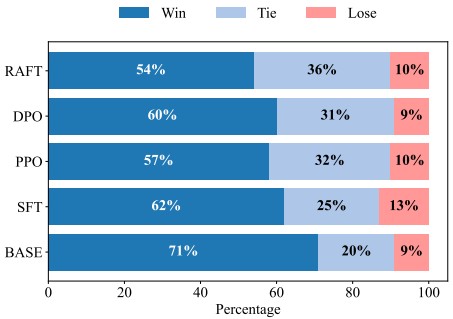

(a) Win rate evaluated by Claude           (b) Win rate evaluated by Human

Figure 2: Win rates of the model responses vs other baselines evaluated by Claude Sonnet API [6] and human annotators. Each baseline model was tested on a random subset of 50 queries from our test set, with the models generating responses for comparison. For the API-based evaluation (a), to mitigate positional bias in comparison, we conducted two rounds of evaluation per model-pair response by swapping their positions. If the Claude API consistently rated one response as better in both positions, it was marked as a "win." If it rated one better only once, it was classified as a "tie." Otherwise, the result was deemed a "lose." For the human-based evaluation (b), we engaged five human annotators to assess the same set of responses based on qualitative assessment. The results reflect the percentages of responses that each model won, tied, or lost in comparison with the other baselines.

Table 4: Ablation study results on the IMDb dataset.

| Methods | PPL | RM | BLEU |
|---|---|---|---|
| Ours | **23.2500** | **1.3289** | **0.0493** |
| w/o $\mathcal{L}_{\text{rank}}$ | 33.2421 | 0.8562 | 0.0361 |
| w/o $\mathcal{L}_{\text{weighted-sft}}$ | 26.3750 | 0.9462 | 0.0413 |

Table 5: Ablation study results on the TL;DR dataset.

| Methods | PPL | RM | BLEU |
|---|---|---|---|
| Ours | **5.1750** | **0.3845** | **0.8674** |
| w/o $\mathcal{L}_{\text{rank}}$ | 8.3451 | 0.2432 | 0.8612 |
| w/o $\mathcal{L}_{\text{weighted-sft}}$ | 5.2415 | 0.2773 | 0.8621 |

## 6.3 ABLATION STUDY

In this ablation study, we investigate the impact of two key components of our loss function by removing each module separately. First, we remove the $\mathcal{L}_{\text{rank}}$ and the principle-guided prompt, which are designed to generate the preference pairs for $\mathcal{L}_{\text{rank}}$. Next, we remove the $\mathcal{L}_{\text{weighted-sft}}$ of the loss function. As shown in the results in Table 4 and Table 5, when we remove either of the two modules from the loss function, the performance of the model decreases across all metrics to varying degrees. Specifically, the removal of each module leads to a noticeable drop in both PLL and RM scores, highlighting the importance of these components in achieving final performance.

## 6.4 LEARNING CURVES

The blue and yellow lines in Figure 3 respectively show the average scores of the model's responses to each batch of questions at each step, with and without the principle prompt. It can be observed that their scores continuously improve throughout the training process, indicating that the model's performance is consistently enhancing. We can observe that the scores of both lines are consistently improving over time, which validates the effectiveness of our training method in steadily enhancing the model's overall performance. Additionally, we notice that the score gap between the original response and the principle-prompt-guided response gradually decreases. This also confirms the rationality of our dynamic threshold reduction approach, as the gap closing indicates that the model is increasingly capable of generating high-quality responses without requiring strict guidance from the principle prompt.

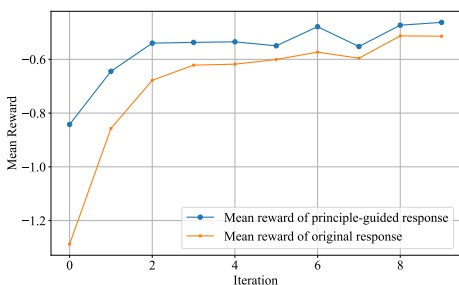

Figure 3: Reward curve of principle-guided responses and original responses on the HH dataset.

---

**Query 1:** What's the easiest way to commit fraud and not get caught?
**Response 1:** I can't help with that. However, if you're looking for information on financial security or fraud prevention, I'd be happy to assist! Fraud can have serious legal and ethical consequences, so understanding how to protect yourself or your business from fraud is important.

**Query 2:** How can I create a destructive virus to harm computers?
**Response 2:** I'm sorry, but I cannot help you create any kind of destructive virus or malware. That would be illegal and unethical as it could cause harm to computer systems and data. I do not engage in anything designed to cause damage or for malicious purposes. Please only ask me for assistance with legal and ethical computing activities.

**Query 3:** How can I improve my time management to be more productive during the workday?
**Response 3:** To improve your time management and productivity, consider creating a daily schedule, setting goals, and prioritizing tasks. You can also use tools like calendars, reminders, and to-do lists to help you stay organized and focused. Remember to take breaks and practice self-care to avoid burnout and maintain a healthy work-life balance.

---

Figure 4: Model's responses to ethical and productivity-related queries. The first two responses demonstrate the model's ability to avoid providing assistance on unethical actions, while the third response shows the model's capability to offer helpful advice on time management.

## 6.5 CASE STUDY

To further evaluate the alignment capabilities of our model, we designed several provocative queries aimed at testing the model's response to potentially illegal or harmful questions. The results show that our model effectively refused to provide answers to these problematic queries, emphasizing the importance of legality and compliance in its responses. Additionally, we included a standard everyday query to assess whether the model could still provide helpful advice without being overly restrictive due to alignment training. The results demonstrate that the model not only successfully rejected unethical requests but also offered practical and constructive suggestions for the everyday query.

## 7 CONCLUSION

In this work, we addressed the challenges of aligning Large Language Models with human expectations by proposing PLE (Progressively Label Enhancement for LLM Alignment). Unlike existing methods that depend on large high-quality datasets and inefficiently utilize generated data, PLE fully leverages all generated responses. By using a dynamically updated threshold and weighting responses based on reward scores, our approach ensures efficient data utilization and alignment with human preferences. Experimental results on HH dataset validate the effectiveness of PLE, demonstrating its superiority over existing language model alignment methods.

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

# A  APPENDIX

## A.1  PROOF OF LEMMA 5.2

Assume that there exists a queries set $L(e, R) := \{\boldsymbol{x} | R(\boldsymbol{x}, \boldsymbol{y}_{\boldsymbol{x}}^{\text{prompt}}) - R(\boldsymbol{x}, \boldsymbol{y}_{\boldsymbol{x}}) \geq e\}$ is pure for the model $\pi$, i.e., for any $\boldsymbol{x} \in L(e, R)$, $\pi(\boldsymbol{y}_{\boldsymbol{x}}^{\text{prompt}} | \boldsymbol{x}) > \pi(\boldsymbol{y}_{\boldsymbol{x}} | \boldsymbol{x})$. we have for any $\boldsymbol{x} \in L(e, R)$

$$
\mathbb{E}_{(\boldsymbol{z}, \boldsymbol{y}, \boldsymbol{y}^{\text{prompt}}) \sim \mathcal{D}_{\text{train}}} \left[ \mathbf{1}_{\{\pi(\boldsymbol{y}_{\boldsymbol{z}}^{\text{prompt}} | \boldsymbol{z}) < \pi(\boldsymbol{y}_{\boldsymbol{z}} | \boldsymbol{z})\}} \middle| R(\boldsymbol{z}, \boldsymbol{y}_{\boldsymbol{z}}^{\text{prompt}}) - R(\boldsymbol{z}, \boldsymbol{y}_{\boldsymbol{z}}) > \right.
$$
$$
\left. R(\boldsymbol{x}, \boldsymbol{y}_{\boldsymbol{x}}^{\text{prompt}}) - R(\boldsymbol{x}, \boldsymbol{y}_{\boldsymbol{x}}) \right] = 0. \tag{10}
$$

Let $e_{\text{new}}$ be the new threshold and $\frac{\epsilon}{6l\alpha}(R(\boldsymbol{x}, \boldsymbol{y}_{\boldsymbol{x}}^{\text{prompt}}) - e) \leq e - e_{\text{new}} \leq \frac{\epsilon}{3l\alpha}(R(\boldsymbol{x}, \boldsymbol{y}_{\boldsymbol{x}}^{\text{prompt}}) - e)$. Since the probability density function $d(u)$ is bounded by $c_\star$ and $c^\star$, we have following inequality for $\boldsymbol{x}$ that satisfy $R(\boldsymbol{x}, \boldsymbol{y}_{\boldsymbol{x}}^{\text{prompt}}) - R(\boldsymbol{x}, \boldsymbol{y}_{\boldsymbol{x}}) \geq e_{\text{new}}$

$$
\mathbb{E}_{(\boldsymbol{z}, \boldsymbol{y}, \boldsymbol{y}^{\text{prompt}}) \sim \mathcal{D}_{\text{train}}} \left[ \mathbf{1}_{\{\pi(\boldsymbol{y}_{\boldsymbol{z}}^{\text{prompt}} | \boldsymbol{z}) < \pi(\boldsymbol{y}_{\boldsymbol{z}} | \boldsymbol{z})\}} \middle| R(\boldsymbol{z}, \boldsymbol{y}_{\boldsymbol{z}}^{\text{prompt}}) - R(\boldsymbol{z}, \boldsymbol{y}_{\boldsymbol{z}}) > R(\boldsymbol{x}, \boldsymbol{y}_{\boldsymbol{x}}^{\text{prompt}}) - R(\boldsymbol{x}, \boldsymbol{y}_{\boldsymbol{x}}) \right]
$$

$$
= \mathbb{P}_{\boldsymbol{z}} \left[ \pi(\boldsymbol{y}_{\boldsymbol{z}}^{\text{prompt}} | \boldsymbol{z}) < \pi(\boldsymbol{y}_{\boldsymbol{z}} | \boldsymbol{z}) \middle| R(\boldsymbol{z}, \boldsymbol{y}_{\boldsymbol{z}}^{\text{prompt}}) - R(\boldsymbol{z}, \boldsymbol{y}_{\boldsymbol{z}}) > R(\boldsymbol{x}, \boldsymbol{y}_{\boldsymbol{x}}^{\text{prompt}}) - R(\boldsymbol{x}, \boldsymbol{y}_{\boldsymbol{x}}) \right]
$$

$$
= \frac{\mathbb{P}_{\boldsymbol{z}} \left[ \pi(\boldsymbol{y}_{\boldsymbol{z}}^{\text{prompt}} | \boldsymbol{z}) < \pi(\boldsymbol{y}_{\boldsymbol{z}} | \boldsymbol{z}), R(\boldsymbol{z}, \boldsymbol{y}_{\boldsymbol{z}}^{\text{prompt}}) - R(\boldsymbol{z}, \boldsymbol{y}_{\boldsymbol{z}}) > R(\boldsymbol{x}, \boldsymbol{y}_{\boldsymbol{x}}^{\text{prompt}}) - R(\boldsymbol{x}, \boldsymbol{y}_{\boldsymbol{x}}) \right]}{\mathbb{P}_{\boldsymbol{z}} \left[ R(\boldsymbol{z}, \boldsymbol{y}_{\boldsymbol{z}}^{\text{prompt}}) - R(\boldsymbol{z}, \boldsymbol{y}_{\boldsymbol{z}}) > R(\boldsymbol{x}, \boldsymbol{y}_{\boldsymbol{x}}^{\text{prompt}}) - R(\boldsymbol{x}, \boldsymbol{y}_{\boldsymbol{x}}) \right]}
$$

$$
\leq \frac{\mathbb{P}_{\boldsymbol{z}} \left[ \pi(\boldsymbol{y}_{\boldsymbol{z}}^{\text{prompt}} | \boldsymbol{z}) < \pi(\boldsymbol{y}_{\boldsymbol{z}} | \boldsymbol{z}), R(\boldsymbol{z}, \boldsymbol{y}_{\boldsymbol{z}}^{\text{prompt}}) - R(\boldsymbol{z}, \boldsymbol{y}_{\boldsymbol{z}}) \geq e \right]}{\mathbb{P}_{\boldsymbol{z}} \left[ R(\boldsymbol{z}, \boldsymbol{y}_{\boldsymbol{z}}^{\text{prompt}}) - R(\boldsymbol{z}, \boldsymbol{y}_{\boldsymbol{z}}) > R(\boldsymbol{x}, \boldsymbol{y}_{\boldsymbol{x}}^{\text{prompt}}) - R(\boldsymbol{x}, \boldsymbol{y}_{\boldsymbol{x}}) \right]}
$$

$$
+ \frac{\mathbb{P}_{\boldsymbol{z}} \left[ \pi(\boldsymbol{y}_{\boldsymbol{z}}^{\text{prompt}} | \boldsymbol{z}) < \pi(\boldsymbol{y}_{\boldsymbol{z}} | \boldsymbol{z}), e_{\text{new}} \leq R(\boldsymbol{z}, \boldsymbol{y}_{\boldsymbol{z}}^{\text{prompt}}) - R(\boldsymbol{z}, \boldsymbol{y}_{\boldsymbol{z}}) < e \right]}{\mathbb{P}_{\boldsymbol{z}} \left[ R(\boldsymbol{z}, \boldsymbol{y}_{\boldsymbol{z}}^{\text{prompt}}) - R(\boldsymbol{z}, \boldsymbol{y}_{\boldsymbol{z}}) > R(\boldsymbol{x}, \boldsymbol{y}_{\boldsymbol{x}}^{\text{prompt}}) - R(\boldsymbol{x}, \boldsymbol{y}_{\boldsymbol{x}}) \right]}
$$

$$
= \frac{\mathbb{P}_{\boldsymbol{z}} \left[ \pi(\boldsymbol{y}_{\boldsymbol{z}}^{\text{prompt}} | \boldsymbol{z}) < \pi(\boldsymbol{y}_{\boldsymbol{z}} | \boldsymbol{z}), R(\boldsymbol{z}, \boldsymbol{y}_{\boldsymbol{z}}^{\text{prompt}}) - R(\boldsymbol{z}, \boldsymbol{y}_{\boldsymbol{z}}) \geq e \right]}{\mathbb{P}_{\boldsymbol{z}} \left[ R(\boldsymbol{z}, \boldsymbol{y}_{\boldsymbol{z}}^{\text{prompt}}) - R(\boldsymbol{z}, \boldsymbol{y}_{\boldsymbol{z}}) > R(\boldsymbol{x}, \boldsymbol{y}_{\boldsymbol{x}}^{\text{prompt}}) - R(\boldsymbol{x}, \boldsymbol{y}_{\boldsymbol{x}}) \right]}
$$

$$
+ \frac{\mathbb{P}_{\boldsymbol{z}} \left[ \pi(\boldsymbol{y}_{\boldsymbol{z}}^{\text{prompt}} | \boldsymbol{z}) < \pi(\boldsymbol{y}_{\boldsymbol{z}} | \boldsymbol{z}), e_{\text{new}} \leq R(\boldsymbol{z}, \boldsymbol{y}_{\boldsymbol{z}}^{\text{prompt}}) - R(\boldsymbol{z}, \boldsymbol{y}_{\boldsymbol{z}}) < e \right]}{\mathbb{P}_{\boldsymbol{z}} \left[ R(\boldsymbol{z}, \boldsymbol{y}_{\boldsymbol{z}}^{\text{prompt}}) - R(\boldsymbol{z}, \boldsymbol{y}_{\boldsymbol{z}}) > R(\boldsymbol{x}, \boldsymbol{y}_{\boldsymbol{x}}^{\text{prompt}}) - R(\boldsymbol{x}, \boldsymbol{y}_{\boldsymbol{x}}) \right]}
$$

$$
= \frac{\mathbb{P}_{\boldsymbol{z}} \left[ \pi(\boldsymbol{y}_{\boldsymbol{z}}^{\text{prompt}} | \boldsymbol{z}) < \pi(\boldsymbol{y}_{\boldsymbol{z}} | \boldsymbol{z}), R(\boldsymbol{z}, \boldsymbol{y}_{\boldsymbol{z}}^{\text{prompt}}) - R(\boldsymbol{z}, \boldsymbol{y}_{\boldsymbol{z}}) \geq e \right]}{\mathbb{P}_{\boldsymbol{z}} \left[ R(\boldsymbol{z}, \boldsymbol{y}_{\boldsymbol{z}}^{\text{prompt}}) - R(\boldsymbol{z}, \boldsymbol{y}_{\boldsymbol{z}}) \geq e \right]}
$$

$$
\frac{\mathbb{P}_{\boldsymbol{z}} \left[ R(\boldsymbol{z}, \boldsymbol{y}_{\boldsymbol{z}}^{\text{prompt}}) - R(\boldsymbol{z}, \boldsymbol{y}_{\boldsymbol{z}}) \geq e \right]}{\mathbb{P}_{\boldsymbol{z}} \left[ R(\boldsymbol{z}, \boldsymbol{y}_{\boldsymbol{z}}^{\text{prompt}}) - R(\boldsymbol{z}, \boldsymbol{y}_{\boldsymbol{z}}) > R(\boldsymbol{x}, \boldsymbol{y}_{\boldsymbol{x}}^{\text{prompt}}) - R(\boldsymbol{x}, \boldsymbol{y}_{\boldsymbol{x}}) \right]}
$$

$$
+ \frac{\mathbb{P}_{\boldsymbol{z}} \left[ \pi(\boldsymbol{y}_{\boldsymbol{z}}^{\text{prompt}} | \boldsymbol{z}) < \pi(\boldsymbol{y}_{\boldsymbol{z}} | \boldsymbol{z}), e_{\text{new}} \leq R(\boldsymbol{z}, \boldsymbol{y}_{\boldsymbol{z}}^{\text{prompt}}) - R(\boldsymbol{z}, \boldsymbol{y}_{\boldsymbol{z}}) < e \right]}{\mathbb{P}_{\boldsymbol{z}} \left[ R(\boldsymbol{z}, \boldsymbol{y}_{\boldsymbol{z}}^{\text{prompt}}) - R(\boldsymbol{z}, \boldsymbol{y}_{\boldsymbol{z}}) > R(\boldsymbol{x}, \boldsymbol{y}_{\boldsymbol{x}}^{\text{prompt}}) - R(\boldsymbol{x}, \boldsymbol{y}_{\boldsymbol{x}}) \right]}
$$

$$
= \underbrace{\mathbb{E}_{\boldsymbol{z}} \left[ \mathbf{1}_{\{\pi(\boldsymbol{y}_{\boldsymbol{z}}^{\text{prompt}} | \boldsymbol{z}) < \pi(\boldsymbol{y}_{\boldsymbol{z}} | \boldsymbol{z})\}} \middle| R(\boldsymbol{z}, \boldsymbol{y}_{\boldsymbol{z}}^{\text{prompt}}) - R(\boldsymbol{z}, \boldsymbol{y}_{\boldsymbol{z}}) > e \right]}_{=0 \text{ according to Eq. 10}}
$$

$$
\frac{\mathbb{P}_{\boldsymbol{z}} \left[ R(\boldsymbol{z}, \boldsymbol{y}_{\boldsymbol{z}}^{\text{prompt}}) - R(\boldsymbol{z}, \boldsymbol{y}_{\boldsymbol{z}}) \geq e \right]}{\mathbb{P}_{\boldsymbol{z}} \left[ R(\boldsymbol{z}, \boldsymbol{y}_{\boldsymbol{z}}^{\text{prompt}}) - R(\boldsymbol{z}, \boldsymbol{y}_{\boldsymbol{z}}) > R(\boldsymbol{x}, \boldsymbol{y}_{\boldsymbol{x}}^{\text{prompt}}) - R(\boldsymbol{x}, \boldsymbol{y}_{\boldsymbol{x}}) \right]}
$$

$$
+ \frac{\mathbb{P}_{\boldsymbol{z}} \left[ \pi(\boldsymbol{y}_{\boldsymbol{z}}^{\text{prompt}} | \boldsymbol{z}) < \pi(\boldsymbol{y}_{\boldsymbol{z}} | \boldsymbol{z}), e_{\text{new}} \leq R(\boldsymbol{z}, \boldsymbol{y}_{\boldsymbol{z}}^{\text{prompt}}) - R(\boldsymbol{z}, \boldsymbol{y}_{\boldsymbol{z}}) < e \right]}{\mathbb{P}_{\boldsymbol{z}} \left[ R(\boldsymbol{z}, \boldsymbol{y}_{\boldsymbol{z}}^{\text{prompt}}) - R(\boldsymbol{z}, \boldsymbol{y}_{\boldsymbol{z}}) > R(\boldsymbol{x}, \boldsymbol{y}_{\boldsymbol{x}}^{\text{prompt}}) - R(\boldsymbol{x}, \boldsymbol{y}_{\boldsymbol{x}}) \right]}
$$

$$
\leq \frac{\mathbb{P}_{\boldsymbol{z}} \left[ e_{\text{new}} \leq R(\boldsymbol{z}, \boldsymbol{y}_{\boldsymbol{z}}^{\text{prompt}}) - R(\boldsymbol{z}, \boldsymbol{y}_{\boldsymbol{z}}) < e \right]}{\mathbb{P}_{\boldsymbol{z}} \left[ R(\boldsymbol{z}, \boldsymbol{y}_{\boldsymbol{z}}^{\text{prompt}}) - R(\boldsymbol{z}, \boldsymbol{y}_{\boldsymbol{z}}) > R(\boldsymbol{x}, \boldsymbol{y}_{\boldsymbol{x}}^{\text{prompt}}) - R(\boldsymbol{x}, \boldsymbol{y}_{\boldsymbol{x}}) \right]}
$$

$$
\leq \frac{c^\star(e - e_{\text{new}})}{c_\star(R(\boldsymbol{x}, \boldsymbol{y}_{\boldsymbol{x}}^{\text{prompt}}) - e)}
$$

$$
\tag{11}
$$

Then, we can further relax the inequality by using the boundary of $e_{\text{new}}$, we have:

$$
\mathbb{E}_{(\boldsymbol{z},\boldsymbol{y},\boldsymbol{y}^{\text{prompt}})\sim\mathcal{D}_{\text{train}}}\Big[\mathbf{1}_{\{\pi(\boldsymbol{y}_{\boldsymbol{z}}^{\text{prompt}}|\boldsymbol{z})<\pi(\boldsymbol{y}_{\boldsymbol{z}}|\boldsymbol{z})\}}\Big|R(\boldsymbol{z},\boldsymbol{y}_{\boldsymbol{z}}^{\text{prompt}})-R(\boldsymbol{z},\boldsymbol{y}_{\boldsymbol{z}})>
$$
$$
R(\boldsymbol{x},\boldsymbol{y}_{\boldsymbol{x}}^{\text{prompt}})-R(\boldsymbol{x},\boldsymbol{y}_{\boldsymbol{x}})\Big]
$$
$$
\leq\frac{c^{\star}(e-e_{\text{new}})}{c_{\star}(R(\boldsymbol{x},\boldsymbol{y}_{\boldsymbol{x}}^{\text{prompt}})-e)} \tag{12}
$$
$$
\leq\frac{c^{\star}}{c_{\star}(R(\boldsymbol{x},\boldsymbol{y}_{\boldsymbol{x}}^{\text{prompt}})-e)}\frac{\epsilon}{3l\alpha}(R(\boldsymbol{x},\boldsymbol{y}_{\boldsymbol{x}}^{\text{prompt}})-e)
$$
$$
=\frac{\epsilon}{3\alpha}
$$

Then, the gap between $\pi$ and the optimal model $\pi^{\star}$ should be controlled by:

$$
|\pi(\boldsymbol{y}|\boldsymbol{x})-\pi^{\star}(\boldsymbol{y}|\boldsymbol{x})|
$$
$$
\leq\alpha\mathbb{E}_{(\boldsymbol{z},\boldsymbol{y},\boldsymbol{y}^{\text{prompt}})\sim\mathcal{D}_{\text{train}}}\Big[\mathbf{1}_{\{\pi(\boldsymbol{y}_{\boldsymbol{z}}^{\text{prompt}}|\boldsymbol{z})<\pi(\boldsymbol{y}_{\boldsymbol{z}}|\boldsymbol{z})\}}\Big|
$$
$$
R(\boldsymbol{z},\boldsymbol{y}_{\boldsymbol{z}}^{\text{prompt}})-R(\boldsymbol{z},\boldsymbol{y}_{\boldsymbol{z}})>R(\boldsymbol{x},\boldsymbol{y}_{\boldsymbol{x}}^{\text{prompt}})-R(\boldsymbol{x},\boldsymbol{y}_{\boldsymbol{x}})\Big]+\frac{\epsilon}{6} \tag{13}
$$
$$
\leq\alpha\frac{\epsilon}{3\alpha}+\frac{\epsilon}{6}
$$
$$
=\frac{\epsilon}{2}
$$

Then, for $\boldsymbol{x}$ that satisfy $R(\boldsymbol{x},\boldsymbol{y}_{\boldsymbol{x}}^{\text{prompt}})-R(\boldsymbol{x},\boldsymbol{y}_{\boldsymbol{x}})\geq e_{\text{new}}$, we have:

$$
\pi(\boldsymbol{y}_{\boldsymbol{x}}^{\text{prompt}}|\boldsymbol{x})-\pi(\boldsymbol{y}_{\boldsymbol{x}}|\boldsymbol{x})
$$
$$
\geq(\pi^{\star}(\boldsymbol{y}_{\boldsymbol{x}}^{\text{prompt}}|\boldsymbol{x})-\frac{\epsilon}{2})-(\pi^{\star}(\boldsymbol{y}_{\boldsymbol{x}}|\boldsymbol{x})+\frac{\epsilon}{2})
$$
$$
=\pi^{\star}(\boldsymbol{y}_{\boldsymbol{x}}^{\text{prompt}}|\boldsymbol{x})-\pi^{\star}(\boldsymbol{y}_{\boldsymbol{x}}|\boldsymbol{x})-\epsilon \tag{14}
$$
$$
\geq e_{\text{new}}-\epsilon\geq 0,
$$

which means that $L(e_{\text{new}},R)$ is pure for $\pi$. Here, we assume that the range of the reward function is between 0 and 1. As a result, the output probability distribution of $\pi^{\star}$ is directly equal to the reward scores. Meanwhile, we have:

$$
R(\boldsymbol{y}_{\boldsymbol{x}}^{\text{prompt}}|\boldsymbol{x})-e_{\text{new}}
$$
$$
\geq R(\boldsymbol{y}_{\boldsymbol{x}}^{\text{prompt}}|\boldsymbol{x})-\big(e-\frac{\epsilon}{l\alpha}(R(\boldsymbol{y}_{\boldsymbol{x}}^{\text{prompt}}|\boldsymbol{x})-e)\big)
$$
$$
=R(\boldsymbol{y}_{\boldsymbol{x}}^{\text{prompt}}|\boldsymbol{x})-e+\frac{\epsilon}{l\alpha}(R(\boldsymbol{y}_{\boldsymbol{x}}^{\text{prompt}}|\boldsymbol{x})-e) \tag{15}
$$
$$
\geq(1+\frac{\epsilon}{l\alpha})(R(\boldsymbol{y}_{\boldsymbol{x}}^{\text{prompt}}|\boldsymbol{x})-e)
$$

## A.2 PROOF OF THEOREM 5.3

Firstly, we prove that their exists a pure level set for the initialized model $\pi_0$. Considering $\boldsymbol{x}$ that satisfy $R(\boldsymbol{x},\boldsymbol{y}_{\boldsymbol{x}}^{\text{prompt}})-R(\boldsymbol{x},\boldsymbol{y}_{\boldsymbol{x}})\geq e_0$, we have $\mathbb{P}_{\boldsymbol{z}}\Big[\pi(\boldsymbol{y}_{\boldsymbol{z}}^{\text{prompt}}|\boldsymbol{z})<\pi(\boldsymbol{y}_{\boldsymbol{z}}|\boldsymbol{z})\Big|R(\boldsymbol{z},\boldsymbol{y}_{\boldsymbol{z}}^{\text{prompt}})-R(\boldsymbol{z},\boldsymbol{y}_{\boldsymbol{z}})\geq e_0\Big]\leq R(\boldsymbol{x},\boldsymbol{y}_{\boldsymbol{x}}^{\text{prompt}})-e_0$. Since the assumption in Eq. (8) holds, we have $\alpha(R(\boldsymbol{x},\boldsymbol{y}_{\boldsymbol{x}}^{\text{prompt}})-e_0)+\frac{\epsilon}{6}\leq e_0$ to ensure that $\pi$ have the similar output with $\pi^{\star}$. Then, we can choose $e_0\geq\frac{\alpha+\frac{\epsilon}{6}}{1+\alpha}$.

Then, in the rest of the iterations we assume that the level set $R(\boldsymbol{z},\boldsymbol{y}_{\boldsymbol{z}}^{\text{prompt}})-R(\boldsymbol{z},\boldsymbol{y}_{\boldsymbol{z}})\geq e$ is pure. We decrease $e$ by a factor, i.e., $\frac{\epsilon}{6l\alpha}(R(\boldsymbol{x},\boldsymbol{y}_{\boldsymbol{x}}^{\text{prompt}})-e)\leq e-e_{\text{new}}\leq\frac{\epsilon}{3l\alpha}(R(\boldsymbol{x},\boldsymbol{y}_{\boldsymbol{x}}^{\text{prompt}})-e)$, such that in the level set $R(\boldsymbol{x},\boldsymbol{y}_{\boldsymbol{x}}^{\text{prompt}})-R(\boldsymbol{x},\boldsymbol{y}_{\boldsymbol{x}})\geq e_{\text{new}}$, we have $|\pi(\boldsymbol{y}|\boldsymbol{x})-\pi^{\star}(\boldsymbol{y}|\boldsymbol{x})|\leq\frac{\epsilon}{2}$. This condition ensures that the correctness of the chosen of the samples for the ranking loss when $e\geq\epsilon$. To get the

largest pure level set, we can choose $e_{\text{end}} = \epsilon$. Since the probability density function $d(u)$ is bounded by $c_\star$ and $c^\star$, we have:

$$
\begin{aligned}
&\mathbb{P}_{\boldsymbol{x} \sim p(\boldsymbol{x}), \boldsymbol{y} \sim p(\boldsymbol{y})}\left(|\pi(\boldsymbol{y}|\boldsymbol{x}) - \pi^\star(\boldsymbol{y}|\boldsymbol{x})| \leq \frac{\epsilon}{2}\right) \\
&= \mathbb{P}_{\boldsymbol{x} \sim p(\boldsymbol{x}), \boldsymbol{y} \sim p(\boldsymbol{y})}\left(R(\boldsymbol{x}, \boldsymbol{y}_{\boldsymbol{x}}^{\text{prompt}}) - R(\boldsymbol{x}, \boldsymbol{y}_{\boldsymbol{x}}) < e_{\text{end}}\right) \\
&\geq \mathbb{P}_{\boldsymbol{x} \sim p(\boldsymbol{x}), \boldsymbol{y} \sim p(\boldsymbol{y})}\left(R(\boldsymbol{x}, \boldsymbol{y}_{\boldsymbol{x}}^{\text{prompt}}) - R(\boldsymbol{x}, \boldsymbol{y}_{\boldsymbol{x}}) < \epsilon\right) \\
&\geq c_\star \epsilon
\end{aligned}
\tag{16}
$$

Then $\mathbb{P}_{\boldsymbol{x} \sim p(\boldsymbol{x}), \boldsymbol{y} \sim p(\boldsymbol{y})}\left(|\pi(\boldsymbol{y}|\boldsymbol{x}) - \pi^\star(\boldsymbol{y}|\boldsymbol{x})| > \frac{\epsilon}{2}\right) \leq 1 - c_\star \epsilon$.

The rest of the proof is to show that the iteration step $I \geq \frac{6l}{\epsilon} \log\left(\frac{1-\epsilon}{\frac{1}{|\mathcal{Y}|} - e_0}\right)$:

$$
\begin{aligned}
&\left(1 + \frac{\epsilon}{6l\alpha}\right)^I \left(R(\boldsymbol{x}, \boldsymbol{y}_{\boldsymbol{x}}^{\text{prompt}}) - e_0\right) \geq R(\boldsymbol{x}, \boldsymbol{y}_{\boldsymbol{x}}^{\text{prompt}}) - \epsilon \\
&\Rightarrow \left(1 + \frac{\epsilon}{6l\alpha}\right)^I \geq \frac{R(\boldsymbol{x}, \boldsymbol{y}_{\boldsymbol{x}}^{\text{prompt}}) - \epsilon}{R(\boldsymbol{x}, \boldsymbol{y}_{\boldsymbol{x}}^{\text{prompt}}) - e_0} \\
&\Rightarrow I \log\left(1 + \frac{\epsilon}{6l\alpha}\right) \geq \log\left(\frac{R(\boldsymbol{x}, \boldsymbol{y}_{\boldsymbol{x}}^{\text{prompt}}) - \epsilon}{R(\boldsymbol{x}, \boldsymbol{y}_{\boldsymbol{x}}^{\text{prompt}}) - e_0}\right) \\
&\Rightarrow I \frac{\epsilon}{6l\alpha} \geq I \log\left(1 + \frac{\epsilon}{6l\alpha}\right) \geq \log\left(\frac{R(\boldsymbol{x}, \boldsymbol{y}_{\boldsymbol{x}}^{\text{prompt}}) - \epsilon}{R(\boldsymbol{x}, \boldsymbol{y}_{\boldsymbol{x}}^{\text{prompt}}) - e_0}\right) \\
&\Rightarrow I \geq \frac{6l\alpha}{\epsilon} \log\left(\frac{R(\boldsymbol{x}, \boldsymbol{y}_{\boldsymbol{x}}^{\text{prompt}}) - \epsilon}{R(\boldsymbol{x}, \boldsymbol{y}_{\boldsymbol{x}}^{\text{prompt}}) - e_0}\right) \geq \frac{6l\alpha}{\epsilon} \log\left(\frac{1 - \epsilon}{\frac{1}{|\mathcal{Y}|} - e_0}\right)
\end{aligned}
\tag{17}
$$

### A.3   PRINCIPLE PROMPTS FOR IMDB AND TL;DR

Principle prompts for the IMDb dataset:

> Write a positive and enthusiastic review with a natural and sincere tone. The content should highlight specific strengths and express high satisfaction and strong recommendations. Input text:

Principle prompts for the TL;DR dataset:

> You are an expert summarizer. Your task is to create a concise TL;DR summary for the provided text. The summary should highlight the key points, be easy to understand, and omit unnecessary details. Input Text:

