# OpenReview forum: "Progressively Label Enhancement for Large Language Model Alignment"
_ICLR.cc/2025/Conference — Submitted to ICLR 2025_

### Official Review · Reviewer_BcLb · 2024-11-01

**Soundness:** 2
**Presentation:** 2
**Contribution:** 2
**Rating:** 3
**Confidence:** 3

**Summary:**

The paper addresses the decoupling of data generation and model training in current RLHF methods by proposing PLE (Progressively Label Enhancement for LLM Alignment). PLE dynamically adjusts the training process based on the quality of the generated data to maximize its utility. Specifically, the contributions of the paper are as follows:

1. The authors are the first to identify that previous alignment methods overlooked the coupling of data generation and training. They introduce PLE to combine these two stages, demonstrating reasonable performance compared to baselines in experiments.

2. The authors theoretically prove that, through a progressively updated threshold strategy, PLE can bound the error rate between the trained model and the optimal model.

**Strengths:**

The authors identify the decoupling issue between data generation and model training in current methods, and propose PLE to improve data utilization. Through both automatic and human evaluation on the HH dataset, they demonstrate that PLE outperforms baseline approaches. The authors theoretically prove that with a progressively updated threshold strategy, PLE can bound the error rate between the trained model and the optimal model.

**Weaknesses:**

1. The authors provide comparative experiments with 4 baselines on the HH dataset and include training curves comparing principle-guided responses with original responses, but I believe the experiments remain insufficient despite the theoretical proof.

- Additional evaluations on datasets like IMDb (as seen in DPO), the Reddit TL;DR summarization dataset, and UltraFeedback would offer valuable insights into model performance on diverse data using PLE.
- It would also be beneficial to test generalization capabilities on out-of-distribution data—for instance, training on UltraFeedback and evaluating on HH.
- Including additional evaluation metrics, such as MSSTR and Distinct-n(as seen in RAFT), could enhance the credibility and robustness of the results.

2. The paper lacks sufficient motivation for introducing progressive approaches and label enhancement methods. Additionally, the experimental results show somewhat minor improvements over the baselines and lack the comprehensiveness needed to justify these components(see weakness 1 for suggestions).

**Questions:**

1. When training baselines like SFT and DPO, it appears that the authors used preference responses from the HH dataset. I’m curious how the baselines would perform if principled guided responses were used as preference responses for training instead.

2. A few potential typos:

    a. For PLE, is the full name “Progressively Label Enhancement for LLM Alignment” as stated on line 21, or “Selective Label Enhancement for Language Model Alignment” as on line 156?

    b. In line 333, “Reward-based Fine FineTuning (RAFT)”—it seems this is not the full name of RAFT.

3. Could the authors expand the experiments on the effects of different components of PLE (e.g., varying loss functions) or hyper-parameters? This would provide deeper insights into which parts contribute most effectively.

4. For Figure 3, how many iterations were trained in total? Are there only 8 iterations in the entire training process?

---

> ### Author Response · Authors · 2024-11-28
>
> We are deeply grateful for your thoughtful review and insightful suggestions. Your feedback has greatly helped us to improve the quality of our manuscript. Regarding your questions, I would like to provide the following explanations:
>
> For Weaknesses:
>
> > 1. The authors provide comparative experiments with 4 baselines on the HH dataset and include training curves comparing principle-guided responses with original responses, but I believe the experiments remain insufficient despite the theoretical proof.
> >
> > - Additional evaluations on datasets like IMDb (as seen in DPO), the Reddit TL;DR summarization dataset, and UltraFeedback would offer valuable insights into model performance on diverse data using PLE.
> > - It would also be beneficial to test generalization capabilities on out-of-distribution data—for instance, training on UltraFeedback and evaluating on HH.
> > - Including additional evaluation metrics, such as MSSTR and Distinct-n(as seen in RAFT), could enhance the credibility and robustness of the results.
>
> In the revised version, we have expanded our experiments to include the Qwen 2.5 model, the IMDb dataset, and the Reddit TL;DR summarization dataset. Additionally, we have introduced BLEU as an evaluation metric and incorporated evaluations using an additional reward model. We also conducted ablation studies to further validate our approach. Due to time constraints, other suggested experiments, such as testing generalization capabilities will be validated in future version.
>
> 2. The paper lacks sufficient motivation for introducing progressive approaches and label enhancement methods. Additionally, the experimental results show somewhat minor improvements over the baselines and lack the comprehensiveness needed to justify these components(see weakness 1 for suggestions).
>
> For Questions:
>
> 1. > When training baselines like SFT and DPO, it appears that the authors used preference responses from the HH dataset. I’m curious how the baselines would perform if principled guided responses were used as preference responses for training instead.
>
>    This is an excellent idea, and we agree that it would provide valuable insights. Due to time constraints, we were unable to include these experiments in the current version, but we plan to explore this in future version.
>
> 2. > A few potential typos:
>    >
>    > a. For PLE, is the full name “Progressively Label Enhancement for LLM Alignment” as stated on line 21, or “Selective Label Enhancement for Language Model Alignment” as on line 156?
>    >
>    > b. In line 333, “Reward-based Fine FineTuning (RAFT)”—it seems this is not the full name of RAFT.
>
>    Thank you for pointing out these typos. We have addressed these typos in the revised version.
>
> 3. > Could the authors expand the experiments on the effects of different components of PLE (e.g., varying loss functions) or hyper-parameters? This would provide deeper insights into which parts contribute most effectively.
>
>    We have added the Ablation Study in section 6.3 in the revised paper. In this ablation study, we investigate the impact of two key components of our loss function by removing each module separately. First, we remove the $ \mathcal{L} _{\text{rank}} $ and the  principle-guided prompt, which are designed to generate the preference pairs for $ \mathcal{L} _{\text{rank}} $. Next, we remove the $ \mathcal{L} _{\text{weighted-sft}} $ of the loss function. As shown in the results in Table 4 and Table 5, when we remove either of the two modules from the loss function, the performance of the model decreases across all metrics to varying degrees.
>
> 4. > For Figure 3, how many iterations were trained in total? Are there only 8 iterations in the entire training process?
>
>    We trained for a total of 10 iterations, generating 1,024 responses in each iteration.
>
> We will incorporate your suggestions into our revisions and hope that they meet your approval. We are open to any further questions or requests for additional information.

---

### Official Review · Reviewer_Ykv4 · 2024-11-02

**Soundness:** 2
**Presentation:** 3
**Contribution:** 2
**Rating:** 5
**Confidence:** 4

**Summary:**

This paper proposes the Progressively Label Enhancement (PLE) framework to improve LLM alignment with human expectations. PLE dynamically adjusts the training process by weighting both principle-guided and original responses based on their reward scores, enhancing data utilization efficiency. Experiment results show that PLE outperforms existing studies.

**Strengths:**

1. The proposed approach is well-motivated.
2. The proposed approach is novel.
3. The approach show good performance.

**Weaknesses:**

1. Although the approach is new, the high-level idea of using prompt to instruct LLM to generated human preferred answer and using reward to optimize the LLM is not new.
2. The paper evaluates model performance using a reward model. It would be better more details of the reward model can be provided. Also, it is possible that the reward model has bias. It would be better that the authors can discuss the effect of the reward model. Otherwise, trying multiple reward models may address the issue.
3. Some details of the experiments are missing. For the results of SFT, PPO, DPO in Table 2, are they the results obtained by the LLM trained by the authors? or they are the results of the instruct version of LLaMA-3 released by Meta? If they are the results obtained from the author, I'm curious about the results of directly evaluating the instruct version of LLaMA-3.

**Questions:**

1. It would be better to make the font size of Figure 1 larger.

---

> ### Author Response · Authors · 2024-11-28
>
> We are deeply grateful for your thoughtful review and insightful suggestions. Your feedback has greatly helped us to improve the quality of our manuscript. Regarding your questions, I would like to provide the following explanations:
>
> For Weaknesses:
>
> 1. > Although the approach is new, the high-level idea of using prompt to instruct LLM to generated human preferred answer and using reward to optimize the LLM is not new.
>
>     Our high-level idea is distinct from merely using prompts to instruct LLMs to generate human-preferred answers. Traditional methods often treat model training and data generation as separate and static processes, which leads to inefficient utilization of the generated data. In contrast, our approach acknowledges the interdependence between these processes, enabling us to fully utilize all samples, regardless of their quality, for efficient and effective model training.
>
> 2. > The paper evaluates model performance using a reward model. It would be better more details of the reward model can be provided. Also, it is possible that the reward model has bias. It would be better that the authors can discuss the effect of the reward model. Otherwise, trying multiple reward models may address the issue.
>
>    The reward model we used is Huggingface RM-Gemma-2B, which has been trained on diverse preference datasets. To address potential biases and provide a more robust evaluation, we also incorporated a larger reward model, Huggingface RM-Mistral-7B, during the evaluation phase in the revised paper.
>
> 3. > Some details of the experiments are missing. For the results of SFT, PPO, DPO in Table 2, are they the results obtained by the LLM trained by the authors? or they are the results of the instruct version of LLaMA-3 released by Meta? If they are the results obtained from the author, I'm curious about the results of directly evaluating the instruct version of LLaMA-3.
>
>    The results of SFT, PPO, and DPO in Table 2 are based on our own training, where we fine-tuned the models using only the HH dataset. We did not compare directly with the instruct version of LLaMA-3 released by Meta, as it is likely fine-tuned on a much larger and diverse set of datasets. Comparing our results with such a model would not be entirely fair, given the differences in training data.
>
> For Questions:
>
> 1. > It would be better to make the font size of Figure 1 larger.
>
>    Thank you for your suggestion. We have increased the font size in Figure 1 in the revised version
>
> We will incorporate your suggestions into our revisions and hope that they meet your approval. We are open to any further questions or requests for additional information.

---

### Official Review · Reviewer_31B5 · 2024-11-04

**Soundness:** 2
**Presentation:** 3
**Contribution:** 2
**Rating:** 5
**Confidence:** 3

**Summary:**

This work proposes a novel language model alignment method by adaptively adjusting the training process based on the generated data quality, which is computed by determining whether a reward score of principle-guided response is higher than the generated response. The authors provide a theoretical analysis to show that the proposed method would lead to a convergence to the optimal model. The experiments are conducted on a single dataset, and the method is shown to achieve a higher performance than the baseline methods (vanilla SFT, DPO, PPO, RAFT).

**Strengths:**

* Proof of the convergence is provided in the work, showing evidence of the effectiveness of the proposed method on the theoretical aspect.

* This paper considers the training process along the data quality together, giving a novel perspective compared to prior alignment works that consider training and data generation as two separate processes.

* The paper is clear and easy to understand.

**Weaknesses:**

* The experiment is only conducted on a single dataset and a single model variant. The paper could be strengthened by a series of post-training analyses.

* Although the method is shown to have better performance on the Helpful and Harmless (HH) dataset, it would be useful if the authors could present a complexity comparison between the baselines, which may seem to be more efficient than the proposed method.

**Questions:**

* Sec 4.1.: `Language model alignment requires a large amount of high-quality data` Consider LIMA, for example, only contains 1k examples, although its base model (65B) is larger than the model used in this work.

* How are the principles integrated into the response generation? How many of such generations are required?

---

> ### Author Response · Authors · 2024-11-28
>
> We are deeply grateful for your thoughtful review and insightful suggestions. Your feedback has greatly helped us to improve the quality of our manuscript. Regarding your questions, I would like to provide the following explanations:
>
> For weaknesses:
>
> 1. > The experiment is only conducted on a single dataset and a single model variant. The paper could be strengthened by a series of post-training analyses.
>
>    We have expanded our experiments to include additional datasets and additional models. Specifically, we added the TLDR dataset for the summarization task, the IMDb dataset for controlled text generation, and Qwen 2.5 7B model for HH dataset. These results are presented in Tables 1, 2 and 3, showcasing the effectiveness of our method across diverse tasks.
>
> 2. > Although the method is shown to have better performance on the Helpful and Harmless (HH) dataset, it would be useful if the authors could present a complexity comparison between the baselines, which may seem to be more efficient than the proposed method.
>
>    Traditional methods treat model training and data generation as separate and static processes, which can lead to inefficient utilization of generated data. In contrast, our method fully leverages all generated samples, avoiding the inefficiencies of data-generation-based methods like RAFT that discard a large portion of generated data. This integrated approach makes our method inherently more efficient while maintaining superior performance.
>
> For Questions:
>
> 1. > Sec 4.1.: `Language model alignment requires a large amount of high-quality data` Consider LIMA, for example, only contains 1k examples, although its base model (65B) is larger than the model used in this work.
>
>    We acknowledge that for simple, everyday tasks, a small amount of alignment data can achieve excellent results, particularly when using sufficiently large base models like LIMA's 65B. However, in most real-world scenarios, more high-quality samples are still necessary. This is especially true for situations where deploying large models is not feasible or for complex tasks such as mathematical reasoning, code generation, or specialized domains like EDA, where a limited number of alignment samples would be insufficient.
>
> 2. > How are the principles integrated into the response generation? How many of such generations are required?
>
>    We integrate the principles into response generation by appending the task-specific prompt to the original input and feeding the combined input to the model. For the Helpful and Harmless (HH) dataset, we conducted 10 iterations, generating 1,024 question-answer pairs in each iteration.
>
> We will incorporate your suggestions into our revisions and hope that they meet your approval. We are open to any further questions or requests for additional information.

---

> > ### Comment · Reviewer_31B5 · 2024-12-02
> >
> > Thanks for the response.
> >
> > Re **Traditional methods treat model training and data generation as separate and static processes, which can lead to inefficient utilization of generated data. In contrast, our method fully leverages all generated samples, avoiding the inefficiencies of data-generation-based methods like RAFT that discard a large portion of generated data. This integrated approach makes our method inherently more efficient while maintaining superior performance.**
> > Without a controlled comparison analysis, I cannot be convinced that the proposed method is more convincing.
> > I will keep the original score.

---

### Official Review · Reviewer_2JFr · 2024-11-04

**Soundness:** 2
**Presentation:** 2
**Contribution:** 2
**Rating:** 5
**Confidence:** 3

**Summary:**

This paper proposed progressively label enhancement for LLM alignment, where the authors prompt the model to generate responses for both the original query and the query guided by a set of carefully designed principles, and then utilize a dynamic threshold to determine the appropriate training approach for both responses based on their corresponding reward scores.
The central idea lies in that at the earlier stage of training, the method tries to use pairs with larger reward margin, as the training proceeds, the reward margin can be decreased.

**Strengths:**

1. prompting with principles is interesting and important;
2. the idea of changing training data progressively using reward function is interesting;
3. there are positive performances in the experiment.

**Weaknesses:**

1. the method does not seem sound to me, there are a few questions left unclear:
1) Essentially, this is a PPO method? so why does EQ 2 have no regularization terms (the KL term between \pi_{sft} and \pi_{\theta};
2) in Eq 7 and the line 15 of the algorithm, the tau seems to be a constant all the time, where you should have \tau_{t+n} = \tau_{t}*\alpha?
3) in the loss functions, normally we are optimizing with mini-batch, which sums all the log probability together in a mini batch (because samples are IID), so a better formulation should be like log\pi_{\theta}? Instead, the authors directly used the probability distribution, which is weird for me.
2. About evaluation:
1) the authors only tested the method with a base model of LLama3 8B, where I believe more base models should be verified, for instance , chatglm-x, qwen-x, and particularly with larger sizes to see if the claims still hold on larger models;
2) the authors only experimented with one dataset, of course more datasets should be validated with;
3) the authors measured the winrate and reward scores over baselines, but it is more interesting to see the results on other benchmarks, for instance, IFEval, MATH, BBQ, and other typical benchmarks.

**Questions:**

See my comments in weaknesses.
note: I did not check into the details the theoretical analysis.
Minor comment:
DPO is not short for direct policy optimization, it is for direct preference optimization.

---

> ### Author Response · Authors · 2024-11-28
>
> We are deeply grateful for your thoughtful review and insightful suggestions. Your feedback has greatly helped us to improve the quality of our manuscript. Regarding your questions, I would like to provide the following explanations:
>
> For Weaknesses:
>
> 1. > Essentially, this is a PPO method? so why does EQ 2 have no regularization terms (the KL term between \pi_{sft} and \pi_{\theta};
>
>    Our method is not based on PPO, and while we do use a reward model to provide feedback scores, the training objective and process are fundamentally different. Eq. 2 represents the ideal scenario where the model seeks to maximize the reward that reflects human values. The KL regularization term, which is used as a trick for training stability in PPO, is not included in the objective function. This choice aligns with the approach [1], where no KL regularization was included to facilitate theoretical analysis.
>
>    [1] Fundamental Limitations of Alignment in Large Language Models, ICML 2024
>
> 2. > in Eq 7 and the line 15 of the algorithm, the tau seems to be a constant all the time, where you should have \tau_{t+n} = \tau_{t}*\alpha?
>
>    Thank you for pointing this out. This is a typo, in the revised version, we have clarified that $\tau$ in Eq 7 and the line 15 of the algorithm, dynamically changes with training iterations.
>
> 3. > in the loss functions, normally we are optimizing with mini-batch, which sums all the log probability together in a mini batch (because samples are IID), so a better formulation should be like log\pi_{\theta}? Instead, the authors directly used the probability distribution, which is weird for me.
>
>    In our implementation, we do use the $\log \pi_\theta$ formulation to optimize the loss function with mini-batches. We have updated the objective function in the revised version to reflect this approach for improved clarity and correctness.
>
> About evaluation:
>
> 1. > the authors only tested the method with a base model of LLama3 8B, where I believe more base models should be verified, for instance , chatglm-x, qwen-x, and particularly with larger sizes to see if the claims still hold on larger models;
>
>    In response, we have included the Qwen 2.5 7B model in our experiments to broaden the evaluation. Due to time constraints during the rebuttal phase, we were unable to test with larger models. However, we plan to incorporate evaluations with larger models in future work to further validate our claims.
>
> 2. > the authors only experimented with one dataset, of course more datasets should be validated with;
>
>    We have expanded our experiments to include additional datasets. Specifically, we added the TLDR dataset for the summarization task and the IMDb dataset for controlled text generation. These results are presented in Tables 2 and 3, showcasing the effectiveness of our method across diverse tasks.
>
> 3. > the authors measured the winrate and reward scores over baselines, but it is more interesting to see the results on other benchmarks, for instance, IFEval, MATH, BBQ, and other typical benchmarks.
>
>    Due to time constraints during the rebuttal phase, we were unable to evaluate our method on these benchmarks. We recognize the value of these benchmarks and plan to include these evaluations in future updates to provide a more comprehensive assessment.
>
> We have incorporated your suggestions into our revisions and hope that they meet your approval. We are open to any further questions or requests for additional information.

---

> > ### Comment · Reviewer_2JFr · 2024-11-28
> >
> > I have read the response. My score keeps unchanged.

---

### Official Review · Reviewer_Xnvs · 2024-11-04

**Soundness:** 2
**Presentation:** 2
**Contribution:** 3
**Rating:** 5
**Confidence:** 4

**Summary:**

The paper proposes a new method for aligning large language models.

The method proposed in the paper is PLE, i.e. Progressively Label Enhancement for LLM Alignment. This framework can dynamically adjust the model's training process based on the evolving quality of the generated data.

Specifically, the method designs a set of principles and uses these principles to guide the model to generate good responses, and then designs a ranking loss and a re-weight loss to train the language models.

On the dataset of HH, the proposed is demonstrated to be better than PPO, and DPO, these commonly used methods.

**Strengths:**

Guiding the language model with a set of principles is a straightforward yet effective approach for generating high-quality responses. This simplicity suggests promising potential for broader application of this method in the future.

**Weaknesses:**

1. **Lack of Ablation Study**: The training loss comprises a ranking loss and a re-weighting loss; however, the paper does not include an ablation study to analyze the individual contributions of these losses to the model’s performance.

2. **Limited Scope of Principle-Based Evaluation**: While using principles proves effective in generating quality responses, the evaluation is limited to alignment tasks. This narrow focus leaves the generalizability of the approach uncertain. Testing across a wider range of tasks, such as mathematical reasoning, summarization, and controlled text generation, would provide stronger evidence of the method’s broader applicability.

3. **Lack of Systematic Study on Principle Design**: The paper does not systematically examine the design of principles or how different principles impact the model’s overall performance.

**Questions:**

Why GPT-4 is not used for annotation when calculating the win rates?

---

> ### Author Response · Authors · 2024-11-28
>
> Thank you for taking the time to review the paper and providing valuable feedback. I appreciate your efforts in ensuring the quality of the research. Regarding your concerns, I would like to provide the following explanations:
>
> For Weaknesses:
>
> 1. > **Lack of Ablation Study**: The training loss comprises a ranking loss and a re-weighting loss; however, the paper does not include an ablation study to analyze the individual contributions of these losses to the model’s performance.
>
>    We have added the Ablation Study in section 6.3 in the revised paper. In this ablation study, we investigate the impact of two key components of our loss function by removing each module separately. First, we remove the $ \mathcal{L} _{\text{rank}} $ and the  principle-guided prompt, which are designed to generate the preference pairs for $ \mathcal{L} _{\text{rank}} $. Next, we remove the $ \mathcal{L} _{\text{weighted-sft}} $ of the loss function. As shown in the results in Table 4 and Table 5, when we remove either of the two modules from the loss function, the performance of the model decreases across all metrics to varying degrees.
>
> 2. > **Limited Scope of Principle-Based Evaluation**: While using principles proves effective in generating quality responses, the evaluation is limited to alignment tasks. This narrow focus leaves the generalizability of the approach uncertain. Testing across a wider range of tasks, such as mathematical reasoning, summarization, and controlled text generation, would provide stronger evidence of the method’s broader applicability.
>
>    We have expanded our evaluation to include additional tasks in the revised paper. Specifically, we have incorporated the TLDR dataset for summarization and the IMDb dataset for controlled text generation, as presented in Tables 2 and 3. The results show that our method demonstrates strong performance across these tasks.
>
> 3. > **Lack of Systematic Study on Principle Design**: The paper does not systematically examine the design of principles or how different principles impact the model’s overall performance.
>
>    Thank you for your insightful comment. Due to time constraints during the rebuttal phase, we were unable to conduct a systematic study on the design of principles. However, we do have a planned approach for future work, where we aim to incorporate specific principle requirements (such as safety) into the design. We plan to evaluate these principles by applying a reward model focused on assessing safety or other princple in the output results. This will allow us to investigate how different principles impact the model's overall performance.
>
> For Questions:
>
> > 1. Why GPT-4 is not used for annotation when calculating the win rates?
>
> We used the Claude model to evaluate win rates, as we believe its performance is comparable to GPT-4 for this task.
>
> We hope that our revisions have addressed all of your concerns, but please let us know if there is anything else we can do to improve the manuscript. We would be happy to answer any additional questions or provide any further information you may need.

---

### Meta-Review · Area_Chair_5S8o · 2024-12-08

**Metareview:**

This paper proposes Progressively Label Enhancement (PLE) for LLM Alignment, which dynamically adjusts the model's training process based on the quality of the generated data. The reviewers raised significant concerns including the novelty of the method and the soundness of the evaluation. No reviewer was willing to support the paper for acceptance.

**Additional Comments On Reviewer Discussion:**

The reviewers reached a consensus on rejection.

---

### Decision · Program_Chairs · 2025-01-22

Reject